# Role of Filter-Feeding Bivalves in the Bioaccumulation and Transmission of White Spot Syndrome Virus (WSSV) in Shrimp Aquaculture Systems

**DOI:** 10.3390/pathogens13121103

**Published:** 2024-12-13

**Authors:** Joon-Gyu Min, Young-Chul Kim, Kwang-Il Kim

**Affiliations:** 1Department of Aquatic Life Medicine, Pukyong National University, Busan 48513, Republic of Korea; cdmin0621@gmail.com; 2Department of Aquatic Life Medicine, Gangneung-Wonju National University, Gangneung 25457, Republic of Korea; sai100@gwnu.ac.kr

**Keywords:** white spot syndrome virus, bioaccumulation, bivalves, viral transmission, filter-feeding

## Abstract

White spot syndrome virus (WSSV) poses a major risk to shrimp aquaculture, and filter-feeding bivalves on shrimp farms may contribute to its persistence and transmission. This study investigated the bioaccumulation and vector potential of WSSV in Pacific oysters (*Crassostrea gigas*), blue mussels (*Mytilus edulis*), and manila clams (*Venerupis philippinarum*) cohabiting with WSSV-infected shrimp. Sixty individuals of each species (average shell lengths: 11.87 cm, 6.97 cm, and 5.7 cm, respectively) cohabitated with WSSV-infected shrimp (*Penaeus vannamei,* average body weight: 16.4 g) for 48 h. In the experiments, bivalves accumulated WSSV particles in both the gill and digestive gland tissues, with the digestive glands exhibiting higher viral load (average viral load, 3.91 × 10^4^ copies/mg), showing that the viral concentrations in bivalve tissues are directly influenced by seawater WSSV concentrations, reaching levels sufficient to induce infection and 100% mortality in healthy shrimp using tissue homogenates. After a 168 h release period in clean water, the WSSV levels in bivalve tissues decreased below the detection thresholds, indicating reduced transmission risk. These results highlight the role of bivalves as temporary reservoirs of WSSV in aquaculture settings, with the transmission risk dependent on the viral concentration and retention period. Our findings suggest that the management of bivalve exposure in WSSV-endemic environments could improve the biosecurity of shrimp farms.

## 1. Introduction

The spread of aquatic diseases is primarily attributed to waterborne transmission and the carriers within the aquatic environment, including vectors and host organisms [1,2,3]. Viral diseases such as the white spot syndrome virus (WSSV) can be transmitted from vectors or carriers to host organisms, highlighting the importance of assessing the risk posed by such carriers to the spread of pathogens [4,5]. WSSV, a highly contagious virus, can cause nearly 100% mortality in shrimp populations within 3–10 days under normal culture conditions, resulting in significant economic losses estimated at USD 8–15 billion globally since its emergence. Since its emergence in the 1990s, WSSV has spread rapidly across the globe, affecting over 50 families of aquatic organisms, including 39 families of crustaceans and 13 non-crustacean species, many of which serve as vectors without showing clinical symptoms [6,7]. The broad host range of WSSV, along with its adaptability to diverse environmental conditions, creates substantial challenges for controlling its spread in shrimp farming and natural aquatic ecosystems. WSSV is capable of transmission through multiple pathways, including waterborne exposure, cohabitation, and trophic interactions, further complicating its management in aquaculture systems [7].

Bivalves such as oysters, mussels, and clams, which filter feed to consume nutrients, can bioaccumulate diverse particles, including viral agents, from their surrounding aquatic environments [8]. Previous studies have reported that viral particles within bivalves attach specifically to the digestive glands and resist inactivation by digestive enzymes, thereby retaining their infectivity [9,10]. Furthermore, viruses accumulated in bivalve tissues can occasionally be released into water under specific conditions, highlighting the potential reinfection of host species and the need to evaluate the role of bivalves in viral transmission in aquatic environments [11,12,13]. Research has shown that not only do bivalves accumulate viral particles from their environment, but the interaction between the viral particles and the digestive processes of the bivalve can also affect viral infectivity, which varies depending on environmental factors, causative agents, and bivalve species [14,15]. Studies such as that by Vazquez-Boucard [16] observed that WSSV DNA was detected in the gills and digestive glands of *Crassostrea gigas* oysters collected from WSD-affected shrimp farms, while oysters from noninfected sites showed no such accumulation. Furthermore, subsequent research by Vazquez-Boucard et al. [17] confirmed that *C. gigas* can accumulate WSSV particles extracellularly in the gills and digestive gland without becoming infected. This indicates that WSSV is likely concentrated through filter-feeding behavior rather than active infection, further supporting the role of oysters as mechanical reservoirs and bioindicator species in aquaculture systems. Additionally, bivalves, exposed to WSSV-contaminated seawater, retained infectivity levels in their tissues, and feeding these WSSV-accumulated bivalves to shrimp resulted in shrimp mortality. These observations indicate that WSSV-accumulated bivalves function as vectors, transmitting the virus and causing significant impacts on shrimp health and survival [18]. Our previous research has shown that WSSV remains more stable in the digestive enzyme environment of bivalves compared to seawater. Such conditions suggest that bivalves may act as reservoirs, increasing the risk of WSSV transmission in aquaculture systems [19]. In addition, our monitoring study revealed that WSSV was detected in 23 of 147 groups (15.6%) of bivalves purchased from domestic markets in Korea, emphasizing the importance of evaluating the potential for WSSV transmission and its impact on aquaculture biosecurity [20].

Understanding the dynamics of WSSV transmission in these environments requires an evaluation of the interactions among hosts, vectors, and environmental factors that influence viral infectivity and persistence. Various environmental conditions, such as organic matter deposition, temperature changes, and microbial interactions, can also reduce the pathogenicity of viruses released into water from infected hosts [12,21,22]. To assess the likelihood of transmission under near-natural conditions, cohabitation models between infected and noninfected species are commonly used to simulate disease spread [1,4,23,24]. The World Organization for Animal Health (WOAH) has highlighted the importance of such natural condition models in identifying susceptible species because they allow for accurate risk assessments of transmission pathways in aquaculture settings [WOAH, Aquatic Animal Health Code]. Consequently, evaluating the risks posed by live bivalves to WSSV transmission through cohabitation with healthy shrimp is crucial, as this can help identify vectors in aquaculture environments.

This study aimed to simulate natural conditions to determine WSSV infectivity levels in bivalves by (i) quantifying WSSV genome copies released from infected shrimp into seawater, (ii) assessing WSSV accumulation and release dynamics within bivalve tissues under cohabitation conditions with infected shrimp, and (iii) determining the vector potential of WSSV-accumulated bivalves through the experimental inoculation of healthy shrimp. The findings of this study may provide insights into the role of bivalves in WSSV transmission in aquaculture systems.

## 2. Materials and Methods

### 2.1. Samples

Whiteleg shrimp (*Penaeus vannamei;* mean body weight, 15.4 ± 2.3 g) were obtained from a whiteleg shrimp farm in Geoje, Gyeongsangnam-do, Republic of Korea. Three major aquaculture species of bivalves in Korea were used for this study: Pacific oyster (*Crassostrea gigas*; mean shell length: 11.87 ± 1.7 cm), blue mussel (*Mytilus edulis*; mean shell length: 6.97 ± 0.4 cm), and manila clam (*Venerupis philippinarum*; mean shell length: 5.7 ± 0.9 cm). The specimens were sourced from the Namcheon beach market in Busan, Republic of Korea. Prior to the experiment, shrimp and bivalves were stocked at a density of 80 individuals per tank to ensure sufficient samples for the experiment while leaving a buffer for potential losses. Each species was acclimated for 7 d in 250 L tanks containing 200 L of seawater maintained at 23 ± 0.5 °C, with complete water change every 3 d. During this period, shrimp were fed twice daily with a commercial diet, and shellfish were fed Reed Mariculture^®^ instant algae shellfish diet 1800 (Reed Mariculture, Campbell, CA, USA) by diluting 50 mL of the product daily with 200 L of seawater, following the manufacturer’s instructions. Five individuals from shrimp and each bivalve species were randomly selected and confirmed to be in a WSSV-free state prior to the experiment using two-step PCR.

For each sampling event, five bivalves per species were immediately dissected to isolate the gill and digestive gland tissues. The tissues were washed twice with PBS (0.1 M, pH 7.2) to remove any external WSSV particles from the surrounding seawater that may have adhered to the surfaces, thereby minimizing contamination. Approximately 10 mg of each tissue was weighed and used for further analysis.

To measure viral release into the seawater, 1 mL of seawater was collected at each sampling time (12 h intervals during experiment), and 200 μL of the sample was used for DNA extraction and subsequent quantitative PCR (qPCR) to determine WSSV genome copy numbers. This allowed for the dynamic monitoring of viral concentrations in seawater over time. The detailed methods for DNA extraction, PCR, and qPCR followed those described in a previous study [20].

### 2.2. Determination of Viral Concentration in Seawater After Inoculation with WSSV Strains of Varying Pathogenicity

Multiple WSSV strains with differing levels of pathogenicity were identified by the previous monitoring of frozen shrimp distributed in the Korean market [20]. From these strains, two were selected for the present study: Kr-1, a highly pathogenic strain, and Kr-4, a strain with relatively lower pathogenicity.

WSSV was inoculated as previously described, with modifications based on a previous study [20]. Briefly, 15 healthy whiteleg shrimp (mean body weight, 16.1 ± 1.7 g) per group were intramuscularly injected with 100 μL of WSSV inoculum containing 1.00 × 10⁵ genome copies per shrimp. The inoculum was prepared by homogenizing the infected abdominal muscle tissue in phosphate-buffered saline (PBS; 1:9, *w*/*v*), centrifuging at 8000× *g* for 10 min, and filtering the supernatant through a 0.45 μm syringe filter to ensure sterility.

Inoculated shrimp were maintained in 100 L tanks at 23 °C, and after the first mortality event, 80% of the tank water was replaced. Dead and moribund shrimp were dissected, pleopods were collected for viral DNA extraction, and WSSV infection was confirmed via PCR analysis.

### 2.3. Cohabitation of WSSV-Infected Shrimp and Bivalves

#### 2.3.1. WSSV Accumulation and Depuration via Cohabitation with Infected Shrimp

The Kr-4 strain, which was found to maintain higher viral concentrations in seawater based on the results presented in Section 2.2, was selected for further experimentation. The whiteleg shrimp (mean body weight, 13.1 ± 2.1 g) were infected with the WSSV Kr-4 strain via an intramuscular injection of 100 μL of inoculum containing 1.00 × 10⁵ genome copies per shrimp (*n* = 30). The shrimp were then maintained under the same conditions described in Section 2.1, and mortality due to WSSV infection was monitored daily for 10 d. Dead and moribund shrimp were immediately removed upon identification.

To investigate the accumulation of WSSV in bivalves, 60 individuals of each bivalve species were introduced into the tank after the first mortality event, allowing them to cohabit with WSSV-infected shrimp (Figure 1). Bivalve sampling for WSSV accumulation was conducted 6, 12, 24, 48, and 72 h post-cohabitation.

To analyze the release of accumulated WSSV from bivalves, individuals were transferred to a fresh 100 L tank when the cumulative mortality rate of the infected shrimp reached 80%. Prior to transfer, the bivalve shells were washed to remove any adhered viral particles and allowed to purge WSSV-contaminated water for 30 min in a new tank before starting the release experiment. Following the transfer, samples were collected from the new tank 12, 24, 72, 120, and 168 h post-transfer. The water in the tank was replaced each time a sample was collected (Figure 1).

#### 2.3.2. Accumulation and Depuration Under High Viral Concentration Cohabitation

To assess WSSV accumulation and release dynamics under relatively high viral concentrations in seawater, an experimental group comprising 60 whiteleg shrimp (mean body weight, 13.1 ± 2.1 g) was artificially infected following the same protocol. The shrimp were injected with 100 μL of WSSV Kr-4 strain inoculum containing 1.00 × 10⁵ genome copies per shrimp and then maintained under the same conditions described in Section 2.1. Mortality due to WSSV infection was observed daily for 10 d, and dead or moribund shrimp were immediately removed.

Following the onset of shrimp mortality, 60 individuals of each bivalve species were introduced to cohabitate with the infected shrimp to evaluate WSSV accumulation (Figure 2). Bivalve samples were collected 12, 24, 36, and 48 h post-cohabitation.

For the release study, sample handling and preparation followed the same procedures described in Section 2.3.1. Viral release from the accumulated WSSV within the bivalve tissues was analyzed by collecting bivalve samples from the tank 3, 6, 12, 24, 72, 120, and 168 h post-transfer. After each sampling, the tank water was completely replaced to maintain consistency (Figure 2).

### 2.4. Assessment of Infectivity of WSSV Accumulated in Bivalve Tissues

To evaluate the pathogenicity of WSSV that accumulate in bivalve tissues, artificial infection trials were conducted using healthy whiteleg shrimp. Digestive gland tissues from each bivalve species that tested positive for WSSV via one-step PCR (after 2 d of cohabitation) were used. For each bivalve type, 100 mg of positive digestive gland tissue was homogenized in 900 μL of PBS (0.1 M, pH 7.2), followed by centrifugation at 8000× *g* for 10 min. The supernatant was then filtered through a 0.45 μm syringe filter to prepare the inoculum.

Five whiteleg shrimp (average body weight, 14.9 ± 2.1 g) were injected with 100 μL of inoculum derived from each bivalve species and maintained in 20 L tanks at 23 °C for a 10-day period to assess infection progression. Additionally, to confirm the elimination of the virus, WSSV-negative tissues—those that underwent a 168 h release period following WSSV accumulation in bivalves and tested negative in PCR—were prepared as inocula and injected into healthy shrimp using the same method.

### 2.5. Statistical Analysis

All graphs were generated using GraphPad Prism software (version 10.0; GraphPad Software, Inc., Boston, MA, USA). Viruses in shellfish were quantified using a two-way analysis of variance with Tukey’s multiple comparison test. Statistical significance was set at *p* < 0.05.

## 3. Results

### 3.1. Viral Concentration in Seawater After Inoculation with WSSV Strains of Varying Pathogenicity

Both the Kr-1 and Kr-4 WSSV strains resulted in 100% cumulative mortality within 6 d post-inoculation. However, the timing of reaching an 80% cumulative mortality rate varied; shrimp inoculated with the Kr-4 strain reached this threshold within 2 d post-inoculation, whereas those inoculated with the Kr-1 strain reached 80% mortality 4 d post-injection (DPI).

Viral genome copies in seawater also differed between the two experimental groups. In the Kr-4 strain group, the highest WSSV concentration was observed 1.5 d post-inoculation at 8.89 × 10⁵ copies/mL, followed by a gradual decline over time. In contrast, shrimp inoculated with the Kr-1 strain exhibited a peak concentration of 8.52 × 10⁶ WSSV genome copies/mL 2 d post-inoculation, with a relatively high concentration maintained over a 3-day period (Figure 3).

### 3.2. Accumulation and Release of WSSV in Bivalves Through Cohabitation with Infected Shrimp

Cumulative mortality and WSSV kinetics in seawater were monitored (Figure 4). The infected shrimp (*n* = 30) had a cumulative mortality rate of 100% at 6 DPI. The concentration of WSSV genome copies released into the water peaked at 1.74 × 10^6^ copies/mL at 3 DPI, when shrimp mortality was 46.7%, and gradually decreased thereafter.

WSSV particles shed by artificially infected shrimp accumulated in the gill and digestive gland tissues of bivalves (60 individuals per species) cohabitating with infected shrimp, resulting in an increase in WSSV genome copies over a 48 h period and confirming viral accumulation (Figure 5). After 48 h of cohabitation, the mean WSSV genome copy concentrations in gill tissues were 1.21 × 10⁴, 1.22 × 10⁴, and 1.59 × 10⁴ copies/mg for oysters, mussels, and clams, respectively. In the digestive gland, the corresponding concentrations were 1.66 × 10⁴, 1.41 × 10⁴, and 7.25 × 10^3^ copies/mg, with clams exhibiting relatively lower accumulation (Table 1).

When 30 bivalves were transferred to fresh seawater 2.5 d post-cohabitation (DPI + 4.5), after reaching an 80.0% cumulative mortality rate, the WSSV concentrations in both gill and digestive gland tissues continuously decreased over the release period. At 120 h post-transfer, all tissue samples had WSSV concentrations below the detection limit of real-time PCR (Figure 6).

### 3.3. Accumulation and Release Under High-Concentration Cohabitation

To investigate WSSV accumulation under relatively high viral concentrations in seawater, 60 whiteleg shrimp were inoculated with WSSV at a dose of 1.00 × 10⁵ copies/shrimp. Mortality began at 2 DPI and reached 100% at 5 DPI. Coinciding with the onset of shrimp mortality, bivalves were introduced into the tank on day 2 to allow for viral accumulation. The WSSV concentration in seawater was 1.73 × 10⁶ genome copies/mL on day 2 and peaked at 2.52 × 10⁷ copies/mL on day 3 (with a shrimp mortality rate of 61.7%). After reaching an 80.0% cumulative mortality rate, the viral titers in the seawater began to decline (Figure 7).

WSSV accumulation in the gill and digestive gland tissues of 60 individuals of each bivalve species was examined during cohabitation with infected shrimp (Table 2). After 48 h, the mean viral concentration in gill tissues was 2.46 × 10^4^, 2.97 × 10^4^, and 2.89 × 10⁴ WSSV genome copies/mg for oysters, mussels, and clams, respectively, showing no significant differences among species. In digestive gland tissues, the corresponding concentrations were 6.40 × 10^4^, 3.74 × 10^4^, and 1.58 × 10^4^ copies/mg, respectively, with clams exhibiting the lowest accumulation levels (Figure 8).

At 2 d post-cohabitation (DPI + 4), when the cumulative shrimp mortality exceeded 80%, the bivalves were transferred to fresh seawater for the release experiment. At 6 h post-transfer, one-step PCR detected no WSSV in any sample, and the detection rates continued to decline over time, indicating ongoing viral release from the bivalve tissues (Table 2). At 168 h post-transfer, both gill and digestive gland tissues from all bivalves exhibited viral concentrations below the real-time PCR detection limit (Figure 9). Notably, mussels showed no detectable levels, even at 120 h post-transfer.

### 3.4. Analysis of Infectivity and Transmission Potential of WSSV Accumulated in Bivalve Tissues

After 2 d of cohabitation with WSSV-infected shrimp in the experiment described in Section 3.3, WSSV genome copies were detected in the digestive gland tissues of oysters, mussels, and clams at concentrations of 2.46 × 10⁴, 2.97 × 10⁴, and 2.89 × 10⁴ copies/mg, respectively. When homogenates from WSSV-accumulated bivalve tissues were injected into healthy whiteleg shrimp, all groups exhibited a cumulative mortality rate of 100% within 10 d. PCR analysis confirmed WSSV infection in all deceased shrimp, indicating mortality due to viral transmission (Figure 10).

In contrast, after a 168 h release period, bivalves that had accumulated WSSV over a 48 h exposure to contaminated seawater had viral levels below the detection limit of 100 genome copies/Rxn. The injection of homogenates from these bivalves into healthy shrimp resulted in no mortality, suggesting that the viral load was insufficient to cause infection.

## 4. Discussion

The role of filter-feeding mollusks, including bivalves, in accumulating environmental contaminants makes them ideal candidates for the biological monitoring of viral and bacterial pollutants in aquatic systems [25,26]. This study builds on previous research by focusing on the potential of bivalves, specifically Pacific oysters (*C. gigas*), blue mussels (*M. edulis*), and manila clams (*V. philippinarum*), as vectors for WSSV in aquaculture environments. Previous studies have largely concentrated on human-related viruses such as norovirus and enteric viruses [9,27], whereas only a limited number of studies have examined bivalves as vectors for aquatic viruses, including VHSV, RSIV, and WSSV [4,23]. The present study extends this knowledge by assessing WSSV accumulation in bivalve tissues and investigating both viral retention and potential infectivity under conditions that simulate shrimp aquaculture.

Our findings indicated that the viral load in the water column significantly affects WSSV accumulation in bivalve tissues. In our cohabitation experiments, WSSV genome copies accumulated in both the gill and digestive gland tissues of the bivalves and peaked within 48 h. Across all bivalve species, gill tissues showed similar concentrations, averaging approximately 1.34 × 10⁴ WSSV genome copies/mg, whereas digestive gland tissues exhibited species-dependent accumulation rates. This pattern suggests that viral bioaccumulation in bivalves is influenced by the tissue type, with the digestive glands potentially offering a more favorable environment for prolonged viral retention owing to the localized concentration of particles ingested through filter-feeding [18]. When comparing species, oysters and mussels had higher viral loads in their digestive glands, whereas clams had approximately half of these levels, although the difference was not significant. These findings highlight species-specific differences likely attributable to variations in filter-feeding rates, tissue structures, and the digestive gland’s capacity for viral retention [14].

In addition, a comparative analysis of shrimp density and WSSV release in a high-concentration cohabitation study further revealed that a higher shrimp density resulted in more pronounced viral accumulation in bivalve tissues. When shrimp density was doubled to 60 individuals in a 200 L tank, the WSSV concentration in seawater reached 2.52 × 10⁷ genome copies/mL by day 3, exceeding levels typically observed in naturally occurring WSD outbreaks in shrimp aquaculture [28]. Bivalves exposed to these high concentrations accumulated WSSV at nearly double the levels observed in the initial experiment, with viral genome copy concentrations in the gill tissues reaching 2.77 × 10⁴ copies/mg. The digestive gland of oysters, mussels, and clams showed even higher accumulation, with viral loads of 6.40 × 10⁴, 3.74 × 10⁴, 1.58 × 10⁴ copies/mg, respectively, suggesting that high seawater viral titers facilitate greater bioaccumulation in bivalves. These results support the hypothesis that the environmental viral load directly influences WSSV bioaccumulation in bivalve tissues, corroborating previous findings on the dependence of viral accumulation on ambient water concentration [11,13]. Moreover, the increased shrimp stocking density is known to elevate interaction frequencies and stress among infected individuals, potentially amplifying viral shedding rates into the environment [29]. This relationship between density and viral shedding aligns with studies demonstrating enhanced WSSV transmission dynamics at higher stocking densities, particularly under conditions favorable to WSSV propagation.

In addition to bioaccumulation, our study evaluated the potential of WSSV-laden bivalves as vectors for transmission. Homogenates from infected bivalve tissues induced 100% shrimp mortality within 10 days, indicating that bivalves exposed to WSSV can serve as effective vectors when integrated into shrimp diets. This result is consistent with earlier studies reporting that even low WSSV genome copy numbers (as few as 10^2^ copies) are sufficient to induce shrimp infection through oral transmission [30,31].

Following the bioaccumulation phase, depuration experiments were conducted to assess the clearance dynamics of WSSV in bivalve tissues. Over a 168 h period in clean water, the WSSV levels in both the gill and digestive gland tissues fell below the PCR detection limit, with Ct values below 100 viral genome copies/Rxn, indicating gradual clearance of the virus from bivalve tissues. In particular, the clearance rates varied among tissues and species; gill tissues generally cleared viral particles faster than the digestive glands, and mussels exhibited more extended viral persistence than oysters and clams.

This aligns with the findings that viruses in bivalve tissues may decrease over time, although some viral particles may persist longer, especially in tissues such as the digestive gland, which processes and retains the ingested material longer [8]. Previous studies indicated that WSSV is more stable in the digestive enzyme environment of bivalves than in seawater, where the virus degrades more rapidly [19]. This suggests that bivalve digestive glands may act as protective reservoirs for WSSV, facilitating its prolonged survival and potential transmission. Our findings support this, as digestive gland tissues showed extended viral persistence compared to gill tissues. Such persistence may have implications for viral transmission in aquaculture, wherein prolonged exposure to high viral loads may result in bivalves serving as reservoirs capable of initiating WSSV outbreaks, even after the initial infection source has been removed. The enhanced stability of WSSV within bivalve digestive systems underscores the importance of understanding their role not only as bioaccumulation vectors but also as amplifiers of viral transmission in aquaculture. Effective management strategies are necessary to mitigate their contribution to the spread of WSSV in aquatic environments.

Our study highlights the importance of managing cohabitation practices in shrimp aquaculture, particularly in WSSV-endemic regions, and underscores the broader implications of these findings for aquaculture management. In addition, bivalves are sometimes integrated into shrimp diets as feed additives [30], increasing the risk of viral transmission through ingestion if the viral load is sufficiently high. Avoiding or carefully controlling bivalve integration in high-risk environments can mitigate WSSV transmission and enhance the biosecurity of shrimp farms. Future studies should investigate the factors affecting WSSV inactivation within bivalve tissues under varying environmental conditions and evaluate the potential of bivalves as long-term biofilters or biocontrol agents in aquaculture systems.

## 5. Conclusions

This study provides new insights into the dynamics of WSSV accumulation and release in bivalves cohabiting with infected shrimp. WSSV accumulates rapidly within bivalve tissues, with concentrations in the digestive glands reaching levels sufficient to induce 100% mortality in healthy shrimp using tissue homogenates. However, following an extended release period in clean water, viral levels in bivalve tissues decreased below the infection thresholds, suggesting that prolonged release can reduce the risk of WSSV transmission from bivalves. These findings highlight the dual role of bivalves as potential vectors and self-limiting reservoirs of WSSV in aquaculture environments, underscoring the need for the controlled use of bivalves in shrimp farms. Our research lays the groundwork for future investigations of WSSV bioaccumulation mechanisms, contributing to the development of safer and more sustainable practices in shrimp aquaculture.

## Figures and Tables

**Figure 1 pathogens-13-01103-f001:**
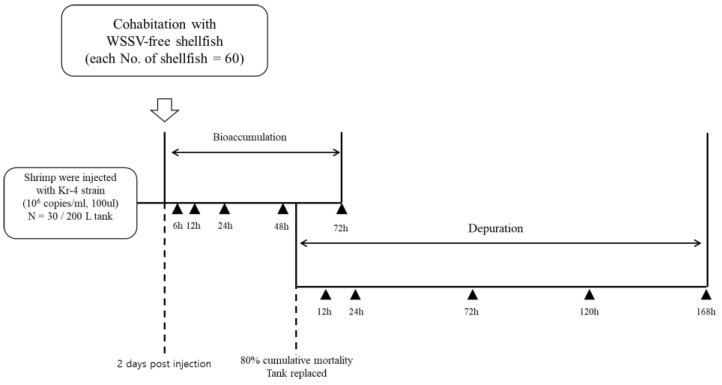
Schematic of the cohabitation of shellfish with WSSV-infected shrimp. Each of 60 shellfish were cohabitated with 30 WSSV-infected shrimp (10^5^ copies/shrimp). Shellfish harboring WSSV particles were transferred to a new 100 L tank for depuration at 23 °C. Sampling times are denoted by black triangles.

**Figure 2 pathogens-13-01103-f002:**
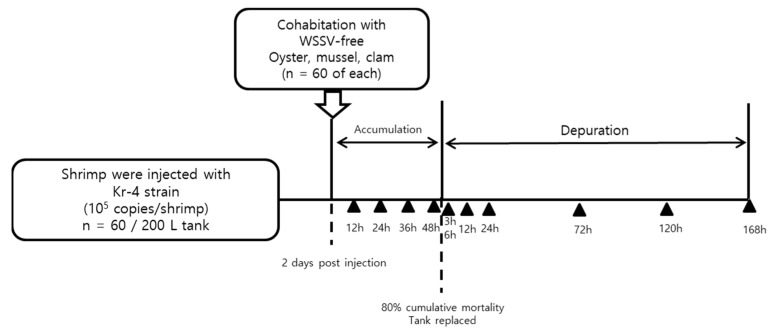
Schematic of the cohabitation of shellfish with WSSV-infected shrimp. Each of 60 shellfish were cohabitated with 60 WSSV-infected shrimp (10^5^ copies/shrimp). Shellfish harboring WSSV particles were transferred to a new 100 L tank for depuration at 23 °C. Sampling times are denoted by black triangles.

**Figure 3 pathogens-13-01103-f003:**
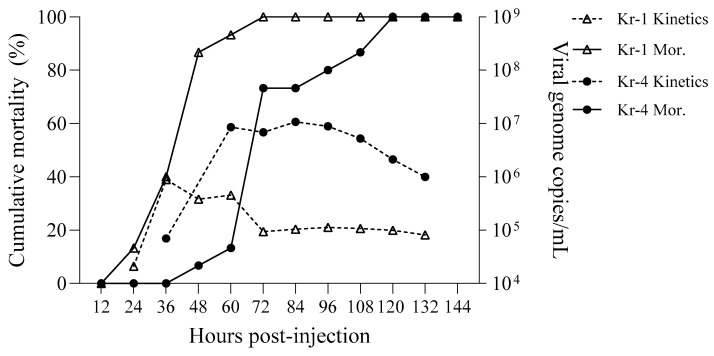
WSSV genome copy numbers in seawater from infected whiteleg shrimp with cumulative mortality (%). The titer of waterborne WSSV (broken line) and daily mortality (line) in tanks injected with the Kr-1 (△) and Kr-4 (●) strains are shown.

**Figure 4 pathogens-13-01103-f004:**
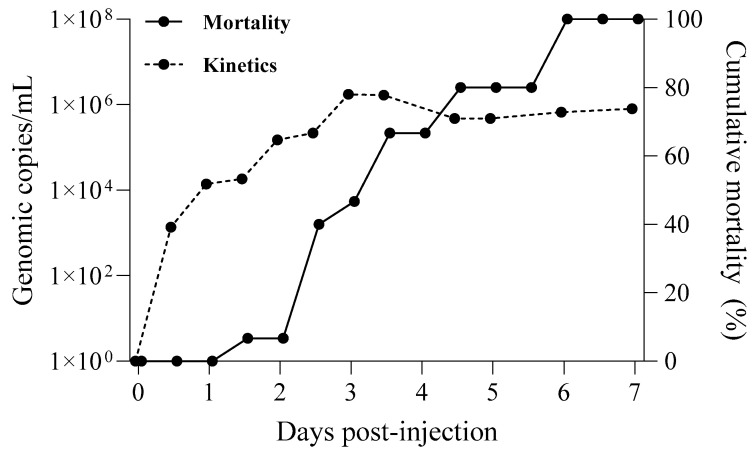
WSSV genome copy numbers in seawater from 30 infected whiteleg shrimp with cumulative mortality (%). The titer of waterborne WSSV (broken line) and daily cumulative mortality (line) in 100 L tanks injected with the Kr-4 strain are shown.

**Figure 5 pathogens-13-01103-f005:**
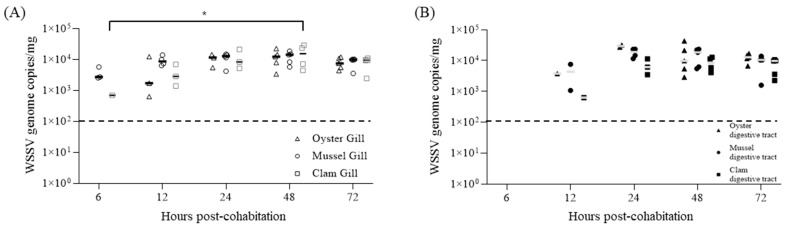
Individual WSSV genome copy numbers in gills (**A**) and digestive glands (**B**) during the low-dose bioaccumulation of oysters (triangle), mussels (circle), and clams (square). Five shellfish were sampled at each time point, and viral copy numbers were determined via qPCR using two-step PCR-positive samples. Horizontal lines indicate the median value, and broken lines indicate the detection limit for all graphs. An asterisk indicates a significant difference (two-way ANOVA, * *p* < 0.05).

**Figure 6 pathogens-13-01103-f006:**
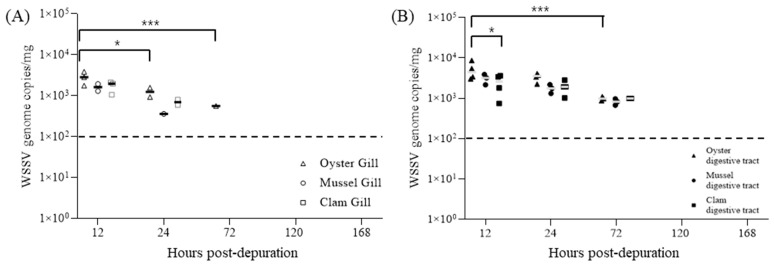
Individual WSSV copy numbers in gills (**A**) and digestive glands (**B**) during the depuration of oysters (triangle), mussels (circle), and clams (square). Five shellfish were sampled at each time point, and viral copy numbers were determined via qPCR using two-step PCR-positive samples. Horizontal lines indicate the median value, and broken lines indicate the detection limit for all graphs. An asterisk indicates significant difference (two-way ANOVA, * *p* < 0.05; *** *p* < 0.001).

**Figure 7 pathogens-13-01103-f007:**
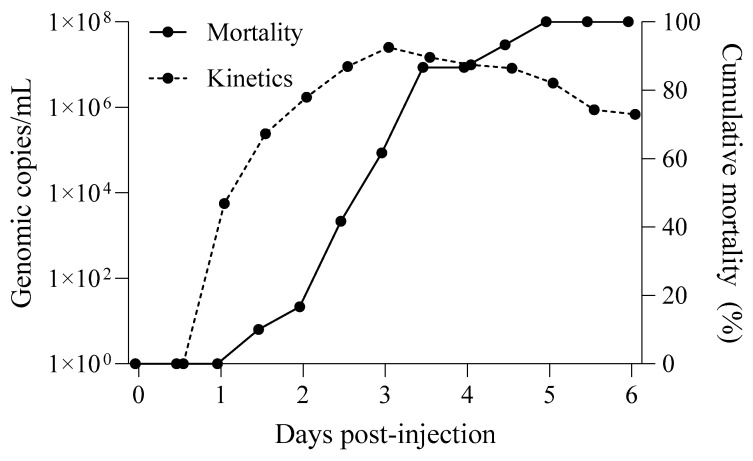
WSSV genome copy numbers in seawater from infected whiteleg shrimp with cumulative mortality (%). The titer of waterborne WSSV (broken line) and daily mortality (line) in tanks injected with the Kr-4 strain are shown.

**Figure 8 pathogens-13-01103-f008:**
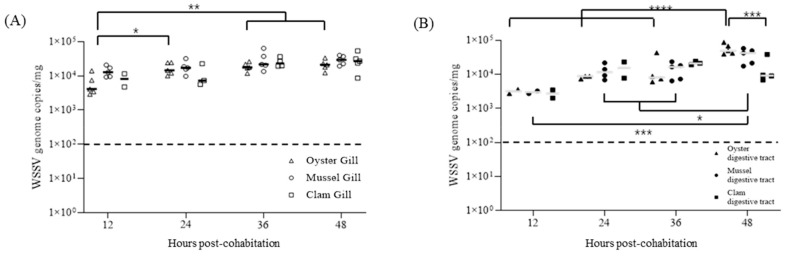
Individual WSSV copy numbers in gills (**A**) and digestive glands (**B**) during the high-dose bioaccumulation of oysters (triangle), mussels (circle), and clams (square). Five shellfish were sampled at each time point, and viral copy numbers were determined via qPCR using two-step PCR-positive samples. Horizontal lines indicate the median value, and broken lines indicate the detection limit for all graphs. An asterisk indicates significant difference (two-way ANOVA, * *p* < 0.05; ** *p* < 0.01; *** *p* < 0.001; **** *p* < 0.0001).

**Figure 9 pathogens-13-01103-f009:**
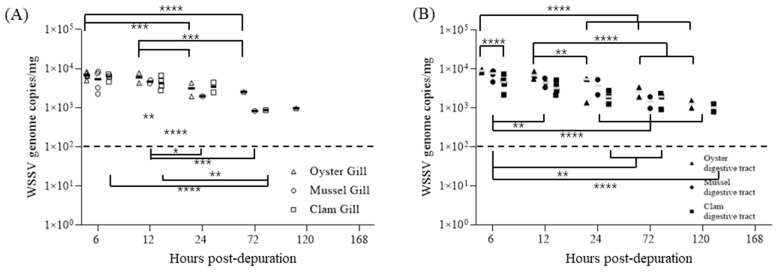
Individual WSSV copy numbers in gills (**A**) and digestive glands (**B**) during the depuration of oysters (triangle), mussels (circle), and clams (square). Five shellfish were sampled at each time point, and viral copy numbers were determined via qPCR using two-step PCR-positive samples. Horizontal lines indicate the median value, and broken lines indicate the detection limit for all graphs. An asterisk indicates significant difference (two-way ANOVA, * *p* < 0.05; ** *p* < 0.01; *** *p* < 0.001; **** *p* < 0.0001).

**Figure 10 pathogens-13-01103-f010:**
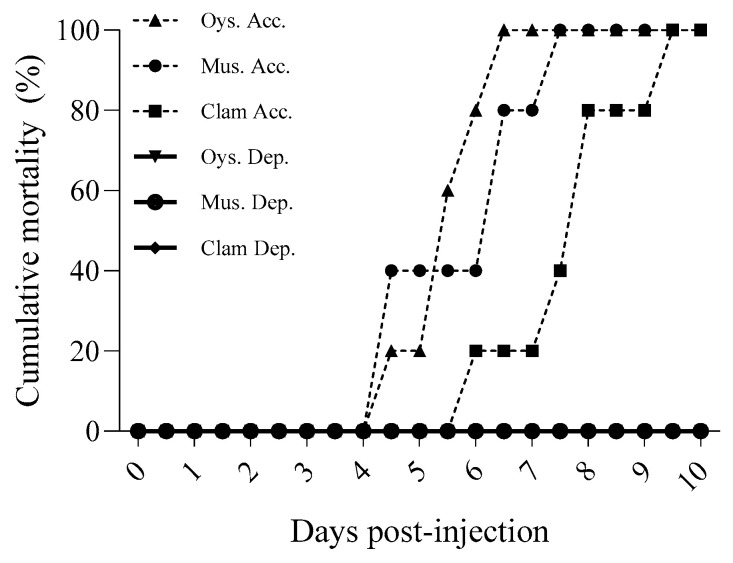
Cumulative mortality (%) of whiteleg shrimp challenged with shellfish-derived WSSV. Digestive gland homogenates from WSSV-accumulated shellfish (2 days of cohabitation with WSSV-infected shrimp) and depurated shellfish (168 h of depuration after accumulation) were intraperitoneally injected (10 mg per shrimp) into five whiteleg shrimp. Symbols represent different shellfish species: oyster (▲), mussel (●), and clam (■). The dashed lines indicate accumulated (Acc.) samples, and the solid lines represent depurated (Dep.) samples.

**Table 1 pathogens-13-01103-t001:** WSSV detection rate in gills and digestive glands of shellfish cohabitating with 30 WSSV-injected shrimp determined using one-step and two-step PCR.

Species of Shellfish	Bioaccumulation Time (h)	No. of Positive Samples/Total Samples Examined			
		One-Step PCR		Two-Step PCR	
		Gill	Digestive Gland	Gill	Digestive Gland
Oyster	6	0/5	0/5	0/5	0/5
	12	1/5	0/5	3/5	1/5
	24	2/5	2/5	3/5	2/5
	48	3/5	3/5	5/5	5/5
	72	2/5	4/5	5/5	5/5
Mussel	6	0/5	0/5	3/5	0/5
	12	2/5	0/5	4/5	2/5
	24	4/5	4/5	5/5	4/5
	48	3/5	3/5	5/5	5/5
	72	4/5	4/5	5/5	5/5
Clam	6	0/5	0/5	1/5	0/5
	12	0/5	0/5	3/5	1/5
	24	1/5	1/5	3/5	3/5
	48	2/5	2/5	4/5	4/5
	72	4/5	3/5	5/5	5/5

**Table 2 pathogens-13-01103-t002:** WSSV detection rate in gills and digestive glands of shellfish cohabitating with 60 WSSV-injected shrimp determined using one-step and two-step PCR.

Species of Shellfish	Bioaccumulation Time (h)	No. of Positive Samples/Total Samples Examined			
		One-Step PCR		Two-Step PCR	
		Gill	Digestive Gland	Gill	Digestive Gland
Oyster	12	1/5	0/5	5/5	2/5
	24	5/5	0/5	5/5	3/5
	36	5/5	1/5	5/5	4/5
	48	5/5	5/5	5/5	5/5
Mussel	12	5/5	0/5	5/5	2/5
	24	5/5	3/5	5/5	4/5
	36	5/5	3/5	5/5	5/5
	48	5/5	5/5	5/5	5/5
Clam	12	1/5	0/5	2/5	2/5
	24	1/5	1/5	3/5	2/5
	36	4/5	3/5	4/5	3/5
	48	4/5	1/5	5/5	4/5

## Data Availability

The original contributions presented in this study are included in the article. Further inquiries can be directed to the corresponding author.

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
