# Peer review of "Role of Filter-Feeding Bivalves in the Bioaccumulation and Transmission of White Spot Syndrome Virus (WSSV) in Shrimp Aquaculture Systems"

_pathogens, 2024, doi:10.3390/pathogens13121103_

Round 1
Reviewer 1 Report
Comments and Suggestions for Authors
Manuscript ID: pathogens-3349180 is properly structured including the chapters required for a scientific article.
From a scientific point of view, we encourage authors to revisit the Materials and methods and Conclusions chapters as follows:
In the Materials and methods chapter, we encourage the authors to present in detail the way the experiments were organized, for example: the number of tanks used, the number of animals populated per tank of both species, the number of days of the experiment, how long after the population the collection took place the samples that were analyzed, the description of the organization of the witness sample, etc., but also other data that the authors consider necessary in this regard.
In the Conclusions chapter, we encourage the authors of the manuscript to develop the conclusions obtained for an added value of the scientific work.
Author Response
Q1. Manuscript ID: pathogens-3349180 is properly structured including the chapters required for a scientific article.
From a scientific point of view, we encourage authors to revisit the Materials and methods and Conclusions chapters as follows:
In the Materials and methods chapter, we encourage the authors to present in detail the way the experiments were organized, for example: the number of tanks used, the number of animals populated per tank of both species, the number of days of the experiment, how long after the population the collection took place the samples that were analyzed, the description of the organization of the witness sample, etc., but also other data that the authors consider necessary in this regard.
In the Conclusions chapter, we encourage the authors of the manuscript to develop the conclusions obtained for an added value of the scientific work.
Response: Thank you for your valuable comments and suggestions for improving our manuscript.
In response to your feedback on the "Materials and Methods" chapter, we have expanded this section to provide detailed descriptions of the experimental design. Specifically, we have included information on the number of animals per tank for each species (L104-107, L149-150, L174), the duration of the experiment, the timeline for sample collection following the population, and the organization of witness samples (L118-123). Additional relevant details supporting the reproducibility of the study have been added to enhance clarity and completeness. Regarding your suggestion for the "Conclusions" chapter, we have revised the text to develop and expand upon the conclusions (L 351-358, L372-410). This revision highlights the key findings of our study and their significance, thus providing added value to the scientific contribution of this work. We believe that these revisions address your comments and improve the quality of the manuscript. Thank you again for your thoughtful suggestion.

Reviewer 2 Report
Comments and Suggestions for Authors
The paper (pathogens-3349180) with the title of "Role of Filter-Feeding Bivalves in the Bioaccumulation and Transmission of White Spot Syndrome Virus (WSSV) in Shrimp Aquaculture Systems" evaluated the alteration and characteristics of WSSV accumulation and release in three bivalves species during cohabitation with infected shrimp. Main content of this manuscript is of high value for the management and biosecurity in aquaculture settings.
However, the presentation of introduction and discussion in this manuscript are less clear and insufficient. The method part lacks of many critical information. Moreover, there are several mistakes in language, syntax, and format.
Therefore, the authors need to modify the corresponding parts of this paper to improve its quality.
Major comments:
1. In the "Abstract" part, multiple critical information on this study is missing, such as the overall experimental design, the initial size of experimental bivalves used for this study, the number of experimental bivalves, the duration of infection, and so on. Therefore, the current "Abstract" part should be clearly written and provided more necessary information.
2. Regarding the "1.Introduction" section, statements on research background were less clear and insufficient. For example, the background information on WSSV, its associated hazards and exposure controls in shrimp aquaculture system should be provided using a concise summary or comparison.
Some descriptions on previous research lacked connection in logic. For example, according to the direct text in Line 42-44, it is difficult to draw a speculative conclusion that WSSV can accumulate in gills and digestive tracts (Line 44-45). It would be preferable to rephrase the corresponding sentence.
Additionally, the text at the beginning of the 3rd paragraph (Line 61- 62) was a little abrupt. This sentence seemed to be more relevant to the main text of the 2nd paragraph. It is suggested that the authors can merge these two parts of similar themes.
Therefore, the authors may rewrite or reorganize the "1.Introduction" part for more clearly specifying background or significance in this study.
3. In the section of "2.Materials and Methods", the overall writing style was too wordy and colloquial, containing several repetitive descriptions. For example, Line 100 and Line 102-103. But multiple methodological descriptions on key information are missing.
First, prior to the experiments, the bivalves are cultured for 7 days to adapt to the laboratory conditions. What about the aquatic environmental parameters (such as salinity, pH, dissolved oxygen, ammonia nitrogen, etc,) under acclimatization condition? These parameters were same as those in the infection/cohabitation experiment or not?
Second, the information on shrimp and bivalves used in this study is incomplete, including the initial body weight and body length, the total number and the number per group, and so on.
Moreover, it is recommended that the authors provide the sample collection in the separate part. Therefore, the authors should add more relevant information. Please revise "2.Materials and Methods" section accordingly.
4. There were multiple unnecessary and redundant descriptions in "3.Results". For example, in 184-187, Line 204-206, etc, the original texts were the methodological descriptions, which have been mentioned in the "2.Materials and Methods" section. Among these descriptions, many were over-long statements. They could be compressed or deleted directly without any negative influence on the corresponding paragraph. Or they could be moved to the "2.Materials and Methods" part.
Additionally, many descriptions in "3.Results" are speculative and discursive. It is suggested to move them to the discussion part. For example, the text of Line 237-238, 280, etc. Similar errors were present in the other parts of "3.Results".
Thus, the main contents of the result part should be simplified and streamlined.
5. Main text of the "4.Discussion" part was not well-organized and inadequate. Multiple statements contain the limited comparisons on the interpretation of results.
The discussion could be more in depth in terms of the similarities/differences of virus bioaccumulation and transmission in bivalve species cultured in aquaculture system between your study and relevant research. It is suggested to rephrase the main text in the discussion part for better emphasizing and clarifying your main findings.
6. The authors should check the references format carefully. The current reference list is a bit chaotic, with volume and page numbers missing, along with other inconsistencies like italic vs. non-italic scientific name of species, capitalized vs. lower-case article titles.
For example, in Reference 2, the information on the journal name, the volume and page number is missing. In many references, the full stop should be provided in the end, such as Reference 1, 7, 10, etc.
In Reference 7, 12, 21, etc, the names of co-authors were incomplete.
Reference2, 10, 21, etc, showed non-italic scientific name of species in article title.
More importantly, the cited literatures published in 2019-2024 are only 7, less than 30% (total literatures: 28). Please make sure about 50% of the references are within 5 years (2019-2024).
Please re-check the reference list carefully and modify accordingly.
Minor comments:
Please check the symbols for volume unit in this study according to the information to related guides. For example, in Figure 4, please replace "ml" with "mL". The authors need to check and revise accordingly.
Other errors (highlighted in yellow) were marked in the PDF file.

There are many unnecessary and redundant descriptions in the main text of this paper. These complicated sentences made the reading difficult. The presentation in the main text, particularly the section of "Introduction" and "Discussion" contained several unclear statements. Also there are still several mistakes, such as the presenting for volume unit and reference list. It is recommended that the text should be proofread by a professional or native speaker.
Author Response
# Reviewer 2
The paper (pathogens-3349180) with the title of "Role of Filter-Feeding Bivalves in the Bioaccumulation and Transmission of White Spot Syndrome Virus (WSSV) in Shrimp Aquaculture Systems" evaluated the alteration and characteristics of WSSV accumulation and release in three bivalves species during cohabitation with infected shrimp. Main content of this manuscript is of high value for the management and biosecurity in aquaculture settings.
However, the presentation of introduction and discussion in this manuscript are less clear and insufficient. The method part lacks of many critical information. Moreover, there are several mistakes in language, syntax, and format.
Therefore, the authors need to modify the corresponding parts of this paper to improve its quality.
Major comments:
- In the "Abstract" part, multiple critical information on this study is missing, such as the overall experimental design, the initial size of experimental bivalves used for this study, the number of experimental bivalves, the duration of infection, and so on. Therefore, the current "Abstract" part should be clearly written and provided more necessary information.
Response: Thank you for your valuable comment. Even though the word count is limited in the Abstract section, as per your comment, we have revised the abstract to more clearly include the most critical information of this study (L17-19).
- Regarding the "1.Introduction" section, statements on research background were less clear and insufficient. For example, the background information on WSSV, its associated hazards and exposure controls in shrimp aquaculture system should be provided using a concise summary or comparison.
Some descriptions on previous research lacked connection in logic. For example, according to the direct text in Line 42-44, it is difficult to draw a speculative conclusion that WSSV can accumulate in gills and digestive tracts (Line 44-45). It would be preferable to rephrase the corresponding sentence.
Additionally, the text at the beginning of the 3rd paragraph (Line 61- 62) was a little abrupt. This sentence seemed to be more relevant to the main text of the 2nd paragraph. It is suggested that the authors can merge these two parts of similar themes.
Therefore, the authors may rewrite or reorganize the "1.Introduction" part for more clearly specifying background or significance in this study.
Response: Thank you for your detailed feedback regarding the "Introduction" section. We have revised this section to provide a clearer and more concise background on WSSV, including its associated hazards and exposure controls in shrimp aquaculture systems. The text at the beginning of the 3rd paragraph (L61–62) in the original MS has been merged with the 2nd paragraph to ensure thematic coherence and smoother transitions (L40-44). Additionally, the sentence in Lines 42–44 has been revised with the inclusion of additional references and rephrased for improved clarity and logical flow, now presented to better support the conclusion about WSSV accumulation in gills and digestive tracts (L56-87). These adjustments aim to enhance clarity and strengthen the logical structure of the "Introduction" section.
- In the section of "2.Materials and Methods", the overall writing style was too wordy and colloquial, containing several repetitive descriptions. For example, Line 100 and Line 102-103. But multiple methodological descriptions on key information are missing.
First, prior to the experiments, the bivalves are cultured for 7 days to adapt to the laboratory conditions. What about the aquatic environmental parameters (such as salinity, pH, dissolved oxygen, ammonia nitrogen, etc,) under acclimatization condition? These parameters were same as those in the infection/cohabitation experiment or not?
Second, the information on shrimp and bivalves used in this study is incomplete, including the initial body weight and body length, the total number and the number per group, and so on.
Moreover, it is recommended that the authors provide the sample collection in the separate part. Therefore, the authors should add more relevant information. Please revise "2.Materials and Methods" section accordingly.
Response: Thank you for your insightful comments regarding the "Materials and Methods" section. As per your suggestion, we have revised Section 2.1 and renamed it as "Samples" to provide a comprehensive overview of the experimental organisms and sample collection process (L 97). In addition, information regarding the experimental organisms, including their body weight, body length, total number, and the number per group, has been added to enhance clarity and completeness (L104-110). Furthermore, details on the environmental conditions and group sizes have been included to address your concerns (L132, 147, 171). However, during the experiment, only salinity was measured, and other environmental parameters, such as pH, dissolved oxygen, and ammonia nitrogen, were not recorded. We acknowledge this limitation but ensured that other experimental conditions, including water source, number of organisms, and temperature, were standardized to match the acclimatization and experimental phases as closely as possible.
- There were multiple unnecessary and redundant descriptions in "3.Results". For example, in 184-187, Line 204-206, etc, the original texts were the methodological descriptions, which have been mentioned in the "2.Materials and Methods" section. Among these descriptions, many were over-long statements. They could be compressed or deleted directly without any negative influence on the corresponding paragraph. Or they could be moved to the "2.Materials and Methods" part.
Additionally, many descriptions in "3.Results" are speculative and discursive. It is suggested to move them to the discussion part. For example, the text of Line 237-238, 280, etc. Similar errors were present in the other parts of "3.Results".
Thus, the main contents of the result part should be simplified and streamlined.
Response: Thank you for your valuable comments regarding the "Results" section. As per your suggestion, we have carefully reviewed and revised the section to address the issues raised. Redundant descriptions and content that overlapped with the "Materials and Methods" section have been deleted to streamline the text. Additionally, speculative and discursive statements have been moved to the "Discussion" section, where they have been rephrased and rewritten to better align with the context of the discussion. These revisions ensure that the "Results" section is more concise and focused, while maintaining clarity and relevance.
- Main text of the "4.Discussion" part was not well-organized and inadequate. Multiple statements contain the limited comparisons on the interpretation of results.
The discussion could be more in depth in terms of the similarities/differences of virus bioaccumulation and transmission in bivalve species cultured in aquaculture system between your study and relevant research. It is suggested to rephrase the main text in the discussion part for better emphasizing and clarifying your main findings.
Response: Thank you for your insightful feedback on the "Discussion" section. As per your suggestion, we have revised the main text to include comparisons with relevant studies, focusing on the similarities and differences in virus bioaccumulation and transmission among bivalve species cultured in aquaculture systems (L351-358, L367-369). Additionally, we have restructured the discussion to better highlight and clarify the main findings of our study while emphasizing its significance in the context of existing research (L372-410).
- The authors should check the references format carefully. The current reference list is a bit chaotic, with volume and page numbers missing, along with other inconsistencies like italic vs. non-italic scientific name of species, capitalized vs. lower-case article titles.
For example, in Reference 2, the information on the journal name, the volume and page number is missing. In many references, the full stop should be provided in the end, such as Reference 1, 7, 10, etc.
In Reference 7, 12, 21, etc, the names of co-authors were incomplete.
Reference2, 10, 21, etc, showed non-italic scientific name of species in article title.
More importantly, the cited literatures published in 2019-2024 are only 7, less than 30% (total literatures: 28). Please make sure about 50% of the references are within 5 years (2019-2024).
Please re-check the reference list carefully and modify accordingly.
Response: Thank you for your detailed comment regarding the reference list. As per your suggestion, we have thoroughly reviewed and revised the references to address all issues raised. Specifically, we ensured consistency of formatting, including proper italicization of scientific names and the inclusion of complete author names, journal names, volume, and page numbers. Additionally, missing punctuation marks have been corrected, and all references have been reformatted according to the required style (L449-520). Regarding the suggestion to include more recent literature (2019–2024), we acknowledge that studies on virus bioaccumulation in bivalves have been limited in recent years. Although we have incorporated additional relevant references within this timeframe, the availability of research directly related to this topic is a limitation. This reflects the current gaps in the field, which we have noted as a limitation of the discussion.
Minor comments:
Please check the symbols for volume unit in this study according to the information to related guides. For example, in Figure 4, please replace "ml" with "mL". The authors need to check and revise accordingly.
Response: Thank you for your comment regarding the volume unit symbols. As per your suggestion, we have reviewed the manuscript and figures thoroughly and replaced all occurrences of "ml" with "mL" to ensure consistency and adherence to the relevant guidelines. These changes have been applied throughout the manuscript, including Figure 4 (L244).

Reviewer 3 Report
Comments and Suggestions for Authors
The paper “Role of Filter-Feeding Bivalves in the Bioaccumulation and 2 Transmission of White Spot Syndrome Virus (WSSV) in 3 Shrimp Aquaculture Systems” has novelty information in the addressed area. However, some minor corrections should be taken into account.
Reviewer’s corrections:
Materials and methods
Line 80. For each species of bivalve mollusk, how many cells of the microalgae mixture were inoculated per milliliter of water daily?
Line 112. How many liters of water did the tank have?
Line 115. Bivalves where fed as above?
Line 156. Please correcto format (105).
Discussion
Line 330. This clam lives buried in the seabed but not in the culture system. Did this affect its filtration capacity?
Line 331. Please, delete the word “statistically”.
References
Line 419. Please, put he species name in italics and check the position of the year (2014).
Lines 422, 423, 435, 461, 468. Please, put he species name in italics.
Line 449. Please put a space before the symbol (&).
Lines 475-476. Please, put Penaeus monodon.

Author Response
# Reviewer 3
The paper “Role of Filter-Feeding Bivalves in the Bioaccumulation and 2 Transmission of White Spot Syndrome Virus (WSSV) in 3 Shrimp Aquaculture Systems” has novelty information in the addressed area. However, some minor corrections should be taken into account.
Q1. Line 80. For each species of bivalve mollusk, how many cells of the microalgae mixture were inoculated per milliliter of water daily?
Response: Thank you for your valuable comment. Considering your comments, we have added the daily feeding amount as follows:
- Providing approximately 10 ml of product (shellfish diet 1800) daily
- In our experiment, we used a daily feed volume of 10ml of shellfish diet 1800 for the 200L tank, based on the soft tissue weight of the 60 individuals from three bivalve species. The total soft tissue weight was estimated 1,200g (C. gigas 10.2g; M. edulis 3.4g; V. philippinarum 6.8g per individuals). Based on the manufacture’s guideline, a daily feed volume of 50 ml shellfish diet 1800 was provided (L108-110).
- Cell Density in the Tank
- With 50 ml of Shellfish Diet 1800 inoculated into the 200 L tank, and given the cell density of the product (~2 billion cells/ml), the resulting cell concentration in the tank was calculated to be approximately 500,000 cells per milliliter.
Q2. Line 112. How many liters of water did the tank have?
Response: Thank you for your question regarding the water volume in the tank. The tank used in our experiment had a total capacity of 250 liters, and we maintained 200 liters of water in the tank to create a consistent aquaria environment for the experiment. This adjustment ensured optimal conditions for the shellfish while aligning with the feeding and experimental parameters. As per your suggestion, we have added the detailed amount of water in the tank in the 2.1 Section (L106-107)
Q3. Line 115. Bivalves where fed as above?
Response: Thank you for your question. The bivalves were fed according to the protocol described in the Methods section, following the manufacturer's guidelines and maintaining consistency throughout the experiment.
Q4. Line 156. Please correcto format (105).
Response: Thank you for your valuable comment. As per your suggestion, we have reviewed (L188).
Q5. Line 330. This clam lives buried in the seabed but not in the culture system. Did this affect its filtration capacity?
Response: Thank you for your insightful question regarding the habitat of Venerupis philippinarum and its potential impact on filtration capacity. The following points were considered in our experimental design:
We hope this explanation clarifies our approach and addresses your concerns.
- philippinarum is known to inhabit seabed sediments (marine substrates), and we acknowledge that this natural habitat may present environmental differences from the experimental setup. These differences could potentially influence filtration capacity.
- However, incorporating sediment into our experimental design posed potential challenges. Specifically, sediments can act as reservoirs for virus accumulation during the cohabitation process. This may lead to:
- i) Chelation of viruses by the sediment, which could alter viral concentration in the seawater and confound experimental results.
- ii) Unintended impacts on the life cycles or behaviors of the other two bivalve species in the study (Crassostrea gigas and Mytilus edulis). To mitigate these risks, sediment was intentionally excluded from the experimental system.
- Recognizing that exposure to bright light could cause stress in philippinarum, which might further influence its filtration capacity, we covered the experimental tanks with curtains to create a dark environment. This adjustment was made to minimize stress and ensure that experimental conditions were as close as possible to natural habitats. As a result, we expected that differences in filtration capacity due to habitat conditions would be minimized.
Q6. Line 331. Please, delete the word “statistically”.
Response: Thank you for your attention to detail. We have removed the word "statistically" as per your suggestion.
Q7. Line 419. Please, put he species name in italics and check the position of the year (2014).
Response: Thank you for your suggestion. As per your suggestion, the species name has been italicized, and the position of the year (2014) has been corrected in the reference (L449).
Q8. Lines 422, 423, 435, 461, 468. Please, put he species name in italics.
Response: Thank you for your comment. As per your comment, all species names have been italicized as per the standard formatting conventions. (L454, 455, 471, 482, 485, 487, 488, 494, 496, 509)
Q9. Line 449. Please put a space before the symbol (&).
Response: Thank you for your comment. As per your suggestion, a space has been added before the symbol (&) to ensure proper formatting. (L 500)
Q10. Lines 475-476. Please, put Penaeus monodon.
Response: Thank you for your comment. As per your comment, the species name Penaeus monodon has been italicized, and a space has been added between the two words to ensure proper formatting. (L 518)

Round 2
Reviewer 2 Report
Comments and Suggestions for Authors
This revised paper (pathogens-3349180) entitled "Role of Filter-Feeding Bivalves in the Bioaccumulation and Transmission of White Spot Syndrome Virus (WSSV) in Shrimp Aquaculture Systems" has been modified in response to the comments of the reviewers. The authors have also addressed the reasons for the unchanged section in the cover letter.
But there are several errors in syntax and format. For example, in Figure 7, please use "mL" instead of the vertical axis of "ml". The full stop is missing in the title of Table 1 and Table 2. Similar errors are present in the end of Figure 8 legend and Figure 9 legend.
Thus, it is recommended to accept this manuscript although there are some minor mistakes.
Author Response
# Reviewer 2
This revised paper (pathogens-3349180) entitled "Role of Filter-Feeding Bivalves in the Bioaccumulation and Transmission of White Spot Syndrome Virus (WSSV) in Shrimp Aquaculture Systems" has been modified in response to the comments of the reviewers. The authors have also addressed the reasons for the unchanged section in the cover letter.
But there are several errors in syntax and format. For example, in Figure 7, please use "mL" instead of the vertical axis of "ml". The full stop is missing in the title of Table 1 and Table 2. Similar errors are present in the end of Figure 8 legend and Figure 9 legend.
Thus, it is recommended to accept this manuscript although there are some minor mistakes.
Response: We sincerely thank the reviewers for their constructive feedback and valuable comments. As per your suggestion, we have corrected the vertical axis unit in Figure 7 from "ml" to "mL" to align with standard guidelines (L243, 277). Additionally, we have added a full stop at the end of the titles for Table 1 and Table 2 (L248, 282) and in the legends of Figure 8 and Figure 9 (L291, 305). These revisions have been implemented throughout the manuscript, and we have carefully reviewed the document to ensure that all similar issues are resolved. We appreciate the reviewers’ valuable insights, which have helped us enhance the quality of our manuscript. Thank you for your consideration.
